# Modulation of the Negative Affective Dimension of Pain: Focus on Selected Neuropeptidergic System Contributions

**DOI:** 10.3390/ijms20164010

**Published:** 2019-08-17

**Authors:** Francesca Felicia Caputi, Laura Rullo, Serena Stamatakos, Sanzio Candeletti, Patrizia Romualdi

**Affiliations:** Department of Pharmacy and Biotechnology, Alma Mater Studiorum-University of Bologna, Irnerio 48, 40126 Bologna, Italy

**Keywords:** pain, negative affect, opioid system, neuropeptides, epigenetics, DNA methylation, DNMT enzymes

## Abstract

It is well known that emotions can interfere with the perception of physical pain, as well as with the development and maintenance of painful conditions. On the other hand, somatic pain can have significant consequences on an individual’s affective behavior. Indeed, pain is defined as a complex and multidimensional experience, which includes both sensory and emotional components, thus exhibiting the features of a highly subjective experience. Over the years, neural pathways involved in the modulation of the different components of pain have been identified, indicating the existence of medial and lateral pain systems, which, respectively, project from medial or lateral thalamic nuclei to reach distinct cortex regions relating to specific functions. However, owing to the limited information concerning how mood state and painful input affect each other, pain treatment is frequently unsatisfactory. Different neuromodulators, including endogenous neuropeptides, appear to be involved in pain-related emotion and in its affective influence on pain perception, thus playing key roles in vulnerability and clinical outcome. Hence, this review article focuses on evidence concerning the modulation of the sensory and affective dimensions of pain, with particular attention given to some selected neuropeptidergic system contributions.

## 1. Introduction

Research conducted over the last 30 years has highlighted that pain, regardless of its nociceptive or neuropathic origin, is a complex and multidimensional experience, incorporating sensory-discriminative, affective-motivational and cognitive-emotional components [1,2]. A coordinated activation of various cerebral areas, defined as the pain neuromatrix, occurs in response to painful input, which, reaching the thalamus (TH) and midbrain structures, promotes long-lasting modifications to neural networks and connections [3]. Notably, the medial pain system projects from the medial thalamic nuclei to the anterior cingulate cortex (ACC), insular cortex and other regions [4,5], where ACC appears to be particularly involved in pain-related emotion and in the motivational aspect of pain [6]. The lateral pain system projects from the lateral thalamic nuclei to the primary and secondary somatosensory cortices (SSC) [7]. This latter circuitry is mainly involved in the sensory-discriminative aspects of pain [8], including the localization and discrimination of pain intensity.

Besides these structures, the prefrontal cortex (PFC) also plays a fundamental role in the control of sensory and affective behaviors [9]. Indeed, the most recent optogenetic studies have shown that the activation of the PFC prelimbic region reduces nociceptive withdrawal latencies, and also decreases aversive and affective responses [10,11], thus producing strong antinociceptive effects. These results are in agreement with other observations, suggesting that the inhibition of PFC projection to the nucleus accumbens (NAc) enhances pain sensitivity [9]. However, PFC may also induce pain chronification via its corticostriatal projection, thus this area could play dual, opposing roles in pain [12]. It has been proposed that PFC-induced pain chronification could depend on the level of dopamine receptor activation along the ventral tegmental area–NAc reward circuitry [12].

In addition to cortical control mechanisms, preclinical and clinical studies clearly showed that other structures, belonging to the brain reward circuit and including the NAc and ventral tegmental area (VTA), also undergo significant functional connectivity alterations during pain states [13]. Moreover, a significant role played by the amygdala (AM) in salience detection and emotional perception has been extensively explored [14,15]. Essentially, the emotional aspects of pain are mainly associated with ACC activation [6], whereas other regions, such as the AM and insula, seem to be mostly involved in the subjective cognitive control of pain [6,16].

Several studies highlighted that different neuromodulators, including endogenous opioid neuropeptides, such as dynorphin (DYN) or nociceptin-orphanin FQ (N/OFQ), as well as the corticotropin-releasing factor (CRF) and the brain-derived neurotrophic factor (BDNF) [17,18,19,20], contribute to the susceptibility and maintenance of chronic pain.

Therefore, in the present review we focused our attention on evidence concerning the modulation of the sensory and affective dimensions of pain, with particular attention to the neuropeptidergic contribution in the aforementioned brain regions.

## 2. Contribution of the Opioidergic System in the Affective Component of Pain

Chronic pain is frequently comorbid with psychiatric disorders and these two pathological conditions often also exhibit dysfunction of the reward mesolimbic circuitry [21,22,23]. In this regard, clinical studies have demonstrated that substance abusers are more likely to develop chronic pain [22,24]. Moreover, a large number of findings have reported that psychiatric disorders are often comorbid with painful conditions [25,26]. For instance, a significant percentage of patients exhibit depressive states in an immediate post-operative period [27], but psychological vulnerability and stress conditions may also lead to worsening and persistent post-surgical pain [28]. On the other hand, positive psychological variables have been reported to influence the long-term outcome with better recovery and a higher quality of life after surgery [29]. These observations point to a relationship between pain and mood disorders and to the fact that these pathological conditions could reinforce each other. Thus, an understanding of the neuromodulation involved in the emotional component of pain becomes crucial for the development of new therapeutic strategies.

In this regard, the well-known ability of the endogenous opioid system to modulate nociception [30,31,32,33], reward [31,32,33] and stress-related behavioral responses [31,34], makes this system a good candidate for a better understanding of these comorbidities.

Endogenous opioids modulate the emotional component of pain through their distinct receptors in different ways [35,36]. The activation of μ-opioid receptors (MORs) represents the gold standard for pain relief [37,38] and is also known to positively modulate mood, which is why their agonists are used for recreational purposes [35,39]. Interesting results were obtained using positron emission tomography (PET) in human volunteers undergoing sustained pain, demonstrating that this condition promotes a regionally specific release of endogenous opioids, which, in turn, causes significant MOR down-regulation [40,41]. In addition, a negative correlation between MOP activation and the sensory and affective scores registered in selected brain regions, including ACC, NAc and AM (McGill Pain Questionnaire subscale), strengthened the crucial role played by MOP in the regulation of pain affect and unpleasantness [40].

The modulation of emotional states by the activation of other opioid receptors is likewise relevant and, in this context, the role of κ -opioid receptors (KORs) in the modulation of emotional states during pain arouses keen interest [35]. Indeed, it is known that, differently from MOR, KOR modulates mood and emotions in contrasting ways [35]. KOR activation mediates the expression of mood disorder symptoms, such as dysphoria in chronic stress conditions [42,43,44], even though systemic KOR agonist administration would seem to exert a solid analgesic effect [45]. In this regard, it is relevant to stress that chronic pain evokes a significant release of dynorphin at spinal level, which seems to contribute to chronic pain maintenance [46,47]. However, other authors suggest a therapeutic potential for κ agonists when their antinociceptive effect is mediated by the absence of arrestin recruitment [48], although KOR supraspinal activation drives anxiety and dysphoria, as has been previously demonstrated [49].

All the discrepancies regarding the role played by the dynorphinergic system in chronic pain conditions, and therefore the question of whether its activation is pro- or anti-nociceptive, has been widely debated [50]. In this context, the different roles exerted by dynorphin are supposed to be mediated through either opioid and/or non-opioid spinal mechanisms [50,51], also taking into account the different contributions of KOR activation at spinal and supraspinal levels [50].

In this regard, recent data produced by our laboratory indicated that neuropathic mice show dynorphinergic system alterations in the corticostriatal circuitry, supporting its role in the negative affective component of pain [18]. Interestingly, we observed a marked prodynorphin (pDYN) gene expression up-regulation in the ACC and PFC, and also significant KOR changes in the same brain regions and in the NAc. Based on our results, we proposed that the up-regulation of pDYN and KOR genes in the PFC could be related to a κ receptor-mediated action of dynorphin. On the other hand, the observed increase of pDYN accompanied by KOR gene expression reduction in the ACC, prompted us to hypothesize that a non-opioid receptor-mediated action occurs in this different cortical area [18]. Accordingly, an NMDA receptor blockade in the ACC prevents the development of neuropathic pain [52].

Through RNAscope multiplex in situ hybridization and Western blotting analysis, the most recent evidence confirms that neuropathic animals exhibit a significant activation of KOR and its endogenous ligand in the NAc, with a specific modulation pattern in males and females [53].

Another crucial aspect in this context is that KOR agonists show lower abuse liability compared to MOR agonists [50,54], probably because KOR activation seems to negatively modulate dopamine release and transmission in the NAc [50,55]. In this view, subsequent studies using a long-acting antagonist confirmed that KOR inhibition restores limbic dopamine release and reward-related behavior [53]. Currently, the possibility of maximizing the analgesic effects and minimizing the dysphoria-like effects is the focus of research applied to the potential use of KOR biased agonists [56].

Although delta-opioid receptors (DORs) were the least studied among classical opioid receptors, in the last decade special attention has also been devoted to understanding their role in pain modulation and emotional disorders. In this view, the efficacy of DOR agonism in the treatment of chronic pain, mainly of inflammatory origin [57,58,59,60], has been well documented. Notably, in an animal model of chronic inflammation due to intraplantar injection of complete Freund’s adjuvant (CFA), the authors reported a significant up-regulation of DOR in the spinal cord [59], which could explain the greater antinociceptive action displayed by DOR agonists in chronic, rather than acute, pain [58,61]. It is also relevant to highlight that DOR seems to modulate emotional responses, since DOR knock-out mice exhibit anxiogenic- and depressive-like behavior [62], thus supporting their relevance to mood disorders [35]. In this context, a specific anxiety and aggression behavior was also observed in pre-proenkephalin deficient mice [63].

According to these results, selected peptide or non-peptide DOR agonists are highly effective in countering negative emotional states [64,65,66]. Taken together, these results suggest a dual usefulness of DOR agonists in chronic pain, for their antinociceptive efficacy and also for their action in the control of negative emotional states associated with persistent pain [67].

The fourth member of the opioid G protein-coupled receptor family, initially termed as opiate-like receptor 1 (ORL-1) [68], was subsequently named nociceptin/orphanin (N/OFQ) peptide receptor (NOP) after its natural ligand N/OFQ was isolated in 1995 [68,69]. A number of studies evaluated the analgesic properties of NOP ligands in different pain conditions, highlighting that NOP-mediated modulation is much more complex than MOR activation [30]. Indeed, different experimental paradigms pointed out that NOP receptor activation could potentially lead to either pronociceptive or antinociceptive effects, depending on the route of administration, dose and experimental pain model [30,70,71].

Different preclinical models of neuropathic pain indicated a series of alterations to the N/OFQ-NOP system in the spinal cord and in dorsal root ganglia [72,73], as well as in selected brain regions involved in pain modulation [17,74]. Investigations conducted in our laboratory observed high levels of N/OFQ peptide in the cerebrospinal fluid of chronic noncancer pain-suffering subjects, whereas a significant reduction of peptide levels was recorded in morphine-treated patients [75].

Moreover, our group recently demonstrated that fourteen days after chronic constriction injury (CCI) mice exhibited a significant increase of nociceptin peptide and NOP mRNA levels in the ACC, but not in the SSC. This result suggests a strong involvement of this peptidergic system in the emotional features of painful experience at supraspinal level [17,76]. Under the same experimental conditions, the hypothesis of a probable N/OFQ control on the descending antinociceptive pathway was supported by a marked increase in N/OFQ peptide content in the AM of injured mice. This modulation could be due to hyperpolarization, driven by N/OFQ upon CRFergic neurons [77]. In this regard, a strict correlation between N/OFQ and CRF has been noted in neuropathic pain [78,79]. Consistent with these studies, Zhang and coworkers demonstrated that NOP receptor antagonists revert allodynia and thermal hyperalgesia induced by traumatic stress, also reducing circulating corticosterone levels [80]. The same authors subsequently reported that NOP knockout mice are protected from the development of stress-mediated allodynia, an effect that, interestingly, was observed in male subjects only [81]. These observations support the contribution of the N/OFQ-NOP system in nociceptive sensitivity, in the emotional component of pain, and also in pain-related hormonal dysregulation.

## 3. CRF and the Endogenous Opioid System: A Reciprocal Connection in Controlling Pain Transmission

Over time, several studies have examined the correlation between CRF and the endogenous opioid system, since they often overlap in the same brain areas, particularly those involved in anxiety and in the affective component of pain. It has been demonstrated that CRF release triggers DYN system activation in the extended AM, increasing glutamate release on the one hand [82] and inhibiting the GABAergic transmission on the other [83]. These pathways can act together to increase anxiety-like behavior [82].

Intriguing evidence highlighted the role of CRF in the affective component of pain. Indeed, an increase in extracellular CRF levels was assessed in the formalin-induced conditioned place aversion (F-CPA) paradigm, in the bed nucleus of stria terminalis (BNST) [19]. Moreover, the infusion of CRF antagonists suppressed the F-CPA but did not alter nociceptive behavior, thus suggesting a prominent role played by CRF in the emotional, rather than sensory, component of pain [19].

An interesting correlation between CRF and the endogenous opioid system has also been obtained from inflammatory pain and immune-derived peripheral antinociception models [84,85]. Indeed, it is known that the release of endogenous opioid peptides by immune cells during inflammatory pain is favored by the action of pro-inflammatory mediators, such as CRF and also interleukin-1β [86,87]. Once released, opioid peptides can activate their high affinity receptors, whose expression on the peripheral ends of sensory neurons is increased during inflammation. Thus, opioids can effectively reduce inflammatory pain [88]. In this regard, a local intraplantar administration of DOR and KOR antagonists in a rat model of inflammatory pain caused a worsening of painful state, thus indicating that the increase of local opioid peptides is involved in peripheral intrinsic analgesia [89].

Another significant correlation between CRF and the opioid system has been observed as regards the descending nociceptive control circuitry. In particular, it seems that the excitability of the central amygdala (CeA) neurons projecting to the periaqueductal gray matter (PAG) (named CeAM-PAG neurons), is strictly dependent on the N/OFQ effect and CRF presence [77], since the N/OFQ perfusion of CeAM-PAG neurons causes the inhibition of the descending pain pathway through CRFergic neuron hyperpolarization [77]. In this context, the marked increase of N/OFQ levels that we observed in the AM, using a CCI model of neuropathic pain, could have a role in the increased nociceptive behavior exhibited by sciatic nerve-injured animals [17].

## 4. BDNF and Opioid Relationship in Pain and the Affective Dimension of Pain

As already mentioned, stress can heavily contribute to chronic pain. In particular, stress, as well as depression or psychological vulnerability, may be due to a long-lasting chronic pain condition and, in turn, these emotional conditions also promote or exacerbate painful states [28,81]. Several studies have shown that stress exposure, as well as mood disorders, may significantly affect nociceptive behavior [90], and may also impair the capacity to modulate pain experience [91]. In this view, special attention has been devoted to the role of the neurotrophin BDNF in neuronal plasticity induced by peripheral inflammation or other pain conditions, as well as in the affective component of pain. In an experimental model of combined stress and injury, a significant up-regulation of BDNF expression was reported in the PFC of rats [92]. Moreover, an overexpression of BDNF has been observed in selected brain regions that are crucial for pain transmission, such as the cortical areas, rostral ventromedial medulla, as well as at spinal level, after nerve injury or during inflammation. In particular, our group recently reported that, 14 days after CCI surgery, mice exhibited neuropathic signs, together with a significant up-regulation of BDNF mRNA levels in the ACC and in PFC [18]. These results are in agreement with the likely pro-nociceptive role of BDNF [93,94,95]. In addition, a significant increase of mature BDNF protein has also been observed in the ACC during inflammatory pain [96].

An interesting correlation between BDNF and DYN alterations, also observed in psychiatric disorders [97], is represented by the significant increase of both genes in the PFC of neuropathic mice, suggesting their likely interaction in the maladaptive neuroplasticity that characterizes neuropathic pain and its affective component [18].

The crucial role of BDNF in the affective-emotional aspect of pain has been well documented by a study showing that the ACC injection of a novel and selective BDNF-tropomyosin receptor kinase B (TrkB) antagonist, cyclotraxin-B, prevented neuronal hyperexcitability, cold hypersensitivity, and passive avoidance behavior [96]. Hence, these results highlighted the important role played by the activation of this neurotrophin signaling in a selected region particularly involved in the emotional dimension of pain. In addition, other research drew attention to the usefulness of TrkB antagonism in counteracting conditioned place preference (CPP) acquisition towards the compartment associated with pain relief, only when the TrkB antagonist was injected into the ACC of nerve-injured rats [95].

Notwithstanding the nociceptive role suggested for BDNF by the above mentioned and other studies [98,99], evidence reporting that after nerve injury the induction of BDNF mRNA and protein in rat ACC is associated with an improvement of pain-related negative emotion also exists [100].

## 5. Analgesic Strategies and Their Ability to Modulate the Affective Dimension of Pain

Opiates represent the “gold standard” in pain control and, indeed, they are the fundamental component of therapeutic strategies for both acute and chronic pain conditions. As described above, endogenous opiates act on different types of opioid receptors, although exogenous drugs mainly target MOR. It has long been known that morphine exerts a stronger effect on the affective than on the sensory dimension of pain [101]. However, the specific neurochemical substrates underlying this modulation have only recently been better defined.

In this regard, LaGraize and colleagues demonstrated that a low dose of systemic morphine injection (0.5 mg/kg) did not alter the mechanical thresholds in neuropathic animals, whereas it attenuated place escape/avoidance behavior. As mentioned, ACC is part of the medial nociceptive system and, indeed, morphine microinjection directly into this cerebral area reduces pain affect but does not alter mechanical paw withdrawal thresholds [102]. These significant results show how the same opiate is able to differently affect the emotional rather than the sensory component of pain, and that this is a MOR-mediated effect, since it can be prevented by systemic naloxone pretreatment.

BNST and CeA represent crucial areas in the regulation of emotional states, also in the light of the relevant interactions among opioid, CRF and neuropeptide Y (NPY) systems, which take place in these brain regions [19,77,82]. Notably, morphine injection in the basolateral amygdala reduces F-CPA, with no effect on nociceptive behavior [103]. Morphine also evokes the same effects when it is administered directly into the ventral part of the BNST (vBNST), an effect that seems to be due to its ability to inhibit neuronal excitability in the type II neurons of vBNST [104,105]. Conversely, CRF directly participates in F-CPA, since it is counteracted by the injection of CRF antagonists into the dorsolateral part of the BNST (dlBNST). Like morphine, NPY exerts an anti-aversive effect with no effect on the sensory component of pain, which would counter CRF activity [19]. Furthermore, Zhang and coworkers demonstrated that the DAMGO injection intra-CeA attenuates the affective component of pain in a much more significant way than the sensory [106].

Basically, these findings substantiate evidence that mood modulation, exerted by MOR activation, also includes positive control of the affective dimension of pain [40] in selected brain regions [103,104].

The involvement of opioid receptors in emotional state modulation is also suggested by some clinical studies investigating atypical antidepressant drugs. It has been highlighted that ketamine, purportedly carrying out its antidepressant action by N-methyl-D-aspartate (NMDA) receptor antagonism, may also act though a mechanism involving opioid receptors [107]. In a randomized double-blinded study, patients resistant to classical antidepressant drugs, exhibited a rapid antidepressant effect after ketamine infusion. Interestingly, naltrexone pre-treatment caused a significant reduction in the ketamine effect [107], thus validating the hypothesis that the ketamine antidepressant effect takes place through opioid receptor activation. This finding emphasizes the relevance of opioid receptor modulation in the control of emotional states, which, as mentioned before, is also a crucial aspect in pain conditions [6,16,25]. In this regard, previous evidence showed that ketamine exerts a kind of central analgesic effect that can be counteracted by MOR and DOR, but not KOR, antagonists [108]. As a matter of fact, the analgesic effect of ketamine has mainly been attributed to NMDA antagonism. However, the engagement of other non NMDA-receptors, including opioids, has also been supported [108,109]. For instance, ketamine administration promotes a lower respiratory depression and spinal/supraspinal analgesia in mice lacking the mu-opioid receptor than in wild-type animals [110]. Moreover, it has also been reported that ketamine may potentiate the analgesic effect of MOR agonists in acute pain conditions [111].

## 6. Epigenetic Mechanisms and Novel Approaches to Controlling the Affective Component of Pain

In mammals, DNA methylation and histone modifications represent the major epigenetic mechanisms implicated in the regulation of gene transcription [112]. DNA methylation is a well-known epigenetic mechanism of gene expression control and is achieved by DNA methyltransferase (DNMT) family enzymes, which include different isoforms (DNMT1, 3a, and 3b) [113]. Through these enzymes, DNA is covalently modified by the addition of methyl groups to the 5-position of the cytosine pyrimidine ring in regions rich in CpG dinucleotides, identified as CpG islands [113]. Mainly, it seems that opiate drugs could increase global DNA methylation levels. In particular, in the leukocytes of patients treated with different opioids, a global increase of DNA methylation has been reported, also including a significant increase of DNA methylation at the MOR promoter region [114]. Interestingly, a positive correlation between the DNA methylation level and pain intensity was described in patients treated with opiates [114]. In this view, a significant up-regulation of DNMT gene expression was reported in an animal model of neuropathic pain [115]. Faced with these facts, we are encouraged to hypothesize that pain conditions, as well as pain intensity, are somehow connected to the degree of DNA methylation.

Currently, genetic and epigenetic changes are generally accepted as likely risk or resilience factors; in this context, pain vulnerability has also been recently associated with the activity of DNMT enzymes. In particular, it is increasingly likely that epigenetic mechanisms, influencing individual variability and susceptibility, could be critically implicated in the pathogenesis of chronic pain [116,117,118]. Epigenetic studies produced significant results in this field. In this regard, mice subjected to partial nerve ligation (PNL) exhibit a significant and long-lasting sensitization of sensory pain compared to sham-operated animals, accompanied by minimal individual variance [119]. On the other hand, the assessment of the affective behavior associated with neuropathic conditions disclosed interesting results. First, PNL animals obviously exhibited a generally higher anxiety-like behavior compared to sham-operated mice. However, results showed that PNL animals were clearly divided into two different groups, vulnerable and resilient to stress. Surprisingly, statistical analysis highlighted no differences between the sham group and PNL stress-resistant mice [119]. Subsequent molecular experiments showed that chronic neuropathy promotes a significant down-regulation of the DNMT3a enzyme isoform in the central nucleus of AM, mainly in vulnerable animals. This evidence suggests that low levels of DNMT3a can induce a worse affective state in painful conditions [119]. This epigenetic approach [119,120] lends support to the classical hypothesis that the emotional experience associated with pain is not only related to pain intensity, but is also significantly affected by individual vulnerability and environmental context [121,122].

The relevance of the DNMT3a isoform has also been confirmed by the observation that peripheral neuropathy evokes up-regulation of the DNMT3a enzyme in dorsal root ganglia (DRG) neurons, whereas DNMT3a knockdown attenuates neuropathic pain signs [123].

Since epigenetic marks are defined as reversible modifications, they can be modulated by pharmacological manipulations [124,125], and therefore could represent useful targets for pain treatment and for its related comorbidities [126]. On this basis, the control of DNA methylation as a pharmacological strategy, together with a deeper knowledge of epigenetic mechanisms, is currently envisaged in several medical fields [126,127,128], including chronic pain [114,117,120,129,130].

## 7. Conclusions

The awareness that pain is a multidimensional experience, characterized by sensory-discriminative, affective-motivational and cognitive-emotional components, offers significant opportunities for the development of new therapeutic strategies aimed at their simultaneous control. In accordance with the evidence reported in this review, the specific neurochemical circuits underlying this complex modulation have been better defined in recent years. In this regard, several studies have highlighted the considerable significance exerted by neuropeptides in the modulation of sensory and emotional pain components in selected brain regions. Moreover, an epigenetic approach might offer new pharmacological tools, not only to manage the affective component of pain, but also to recognize individual vulnerability and environmental factors capable of promoting or exacerbating painful states.

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
