# Peer review of "Modulation of the Negative Affective Dimension of Pain: Focus on Selected Neuropeptidergic System Contributions"

_ijms, 2019, doi:10.3390/ijms20164010_

Round 1

Reviewer 1 Report

In the present review "Modulation of the negative affective dimension of pain: focus on neuropeptidergic contribution", Caputi et al., present a long list of studies that support the role of endogenous opioid system, CRF and BDNF in the modulation of pain at spinal and supraspinal level. Epigenetic mechanisms of pain control are also mentioned at the end of the review. The review has certain merits because it touches on an interesting topic about the mechanisms and treatments of the affective component of pain. However, there are concerns with the review that must be addressed in a revised version. Address

Major concerns:   

1. The title is very broad and somewhat misleading because the review focuses closely only on the endogenous opioids, CRF and BDNF but it does not mention any of the other great number of neuropeptides that modulate different aspects of pain.

2. The authors provide one-sided interpretations of the quoted literature without any discussion of possible alternative observations or data that might contradict their speculations. For example, while some studies assign analgesic functions of the PFC, other demonstrate exactly the opposite, activation of the PFC causes anxiety and pain.  

3. The review suffers of inconsistent approach in the presentation of the various neuropeptidergic systems. While the endogenous opioid system is reviewed by describing in length the function of opioid receptors and use of opioid ligands in pain treatment, CRF receptors are not mentioned at all in the CRF discussion.  

4. The authors claim that …"we focused our attention to the most recent evidence…" however a great number of the quoted papers in the review are 15 to 20 years old.

5. The paragraph about k-opioid receptors (line 87 to 96) is utterly confusing, what is the net effect of k-opioid receptor activation, analgesia or pain?

6. The authors promised that the review is focused on up to date evidence but it is not clear what exactly is the novelty of treatment strategies that are based on a paper published in 1985 (Reference 101).

7. It is puzzling how the last paragraph about epigenetics is in any way connected or relevant to the main body of the manuscript.

8. A consistent problem with the review is the one-sided interpretation of the published literature. The paragraph about BDNF illustrates very well this point. The authors claim that the BDNF function is restricted to the control of the affective component of pain despite the fact that, according to their own words, injection of BDNF antagonists into the ACC prevents cold hypersensitivity, which means that the BDNF also modulates the sensory component of pain. 

Author Response

Responses to Reviewers

Ref: Ms. No. ijms-564777

Title: “Modulation of the negative affective dimension of pain: focus on selected neuropeptidergic system contribution

Reviewer #1

In the present review "Modulation of the negative affective dimension of pain: focus on neuropeptidergic contribution", Caputi et al., present a long list of studies that support the role of endogenous opioid system, CRF and BDNF in the modulation of pain at spinal and supraspinal level. Epigenetic mechanisms of pain control are also mentioned at the end of the review. The review has certain merits because it touches on an interesting topic about the mechanisms and treatments of the affective component of pain. However, there are concerns with the review that must be addressed in a revised version. Address

Major concerns:

Q.1 1. The title is very broad and somewhat misleading because the review focuses closely only on the endogenous opioids, CRF and BDNF but it does not mention any of the other great number of neuropeptides that modulate different aspects of pain.

A.1 We agree with the Reviewer comment and we changed the title, accordingly. To this end, we added the word “selected” in order to narrow the field of the discussed systems.

Q.2 The authors provide one-sided interpretations of the quoted literature without any discussion of possible alternative observations or data that might contradict their speculations. For example, while some studies assign analgesic functions of the PFC, other demonstrate exactly the opposite, activation of the PFC causes anxiety and pain.

A.2 We agree with the Reviewer’s comment  and we changed the text accordingly. In fact , concerning PFC, we added some sentences about a dual role of this brain area mentioning either its antinociceptive or nociceptive significance in pain (see page 2, lines 49-52)

Q.3. The review suffers of inconsistent approach in the presentation of the various neuropeptidergic systems. While the endogenous opioid system is reviewed by describing in length the function of opioid receptors and use of opioid ligands in pain treatment, CRF receptors are not mentioned at all in the CRF discussion.  

A.3 We understand the point raised by Reviewer 1, however the title of paragraph 3 refers only to the interactions between CRF and the endogenous opioid system, without claiming to extensively discuss the CRF system neurobiology.

Q.4 The authors claim that …"we focused our attention to the most recent evidence…" however a great number of the quoted papers in the review are 15 to 20 years old.

A.4

We amended the text of the mentioned phrase: “…we focused our attention to the most recent evidence…", however, we would like to underline that the debate on the relevance of the affective dimension of pain experience has spanned the last 20 years. For this reason we decided to cite, together with recent paper dealing with this topic, also old, anyway relevant, data.

Q.5  The paragraph about k-opioid receptors (line 87 to 96) is utterly confusing, what is the net effect of k-opioid receptor activation, analgesia or pain?

 A.5 We agree with the Reviewer and we thank for the comment.

As a matter of fact, we did some typo errors, so omitting an entire line that now has been correctly repositioned. Now the sentence is clearer and we apologize for the mistake.

Furthermore, we have also to take into account that kappa opioid receptors can play different roles at different brain levels, hence their net role in pain is not so clearcut to be defined.

Q.6 The authors promised that the review is focused on up to date evidence but it is not clear what exactly is the novelty of treatment strategies that are based on a paper published in 1985 (Reference 101).

A.6 We understand the reviewer surprise for citing old papers, such as Ref 101.

However, we simply aimed to underline that it is known for a long time that clinical practice always cited patient’s report revealing the presence of pain after opioid treatment even if less bothersome. Thus, psychic and physical analysis of opiate analgesic action has been of interest since the ‘80s.

Q7 . It is puzzling how the last paragraph about epigenetics is in any way connected or relevant to the main body of the manuscript.

A7

We thank the Reviewer for pointing out this aspect. However, the topic of this review is related to peptidergic neuromodulators. For this reason, we believe that the inclusion of some information on the post-translational mechanisms of neuropeptide expression, that is epigenetic phenomena, could be useful for the reader.

Q8  A consistent problem with the review is the one-sided interpretation of the published literature. The paragraph about BDNF illustrates very well this point. The authors claim that the BDNF function is restricted to the control of the affective component of pain despite the fact that, according to their own words, injection of BDNF antagonists into the ACC prevents cold hypersensitivity, which means that the BDNF also modulates the sensory component of pain. 

A8

We thank the Reviewer for this comment and we apologize for the possible confusing language.

We did not aimed to claim that the BDNF function is restricted to the control of the affective component of pain; in fact, we also mentioned the BDNF modulation of the sensory component of pain. Therefore, we amended the text, accordingly, rephrasing some parts of the paragraph 4.

Reviewer 2 Report

In the proposed review titled “Modulation of the negative affective dimension of pain: focus on neuropeptidergic contribution” the Authors attempt to give an overview on the mechanisms at the basis of chronic pain and the related negative-affective states with particular emphasis on the involvement of neuropeptidergic system. The topic of the review is interesting and overall the manuscript is well organized, thus in my opinion the work is suitable for publication.

Author Response

Responses to Reviewers

Ref: Ms. No. ijms-564777

Title: “Modulation of the negative affective dimension of pain: focus on selected neuropeptidergic system contribution

Reviewer #2

In the proposed review titled “Modulation of the negative affective dimension of pain: focus on neuropeptidergic contribution” the Authors attempt to give an overview on the mechanisms at the basis of chronic pain and the related negative-affective states with particular emphasis on the involvement of neuropeptidergic system. The topic of the review is interesting and overall the manuscript is well organized, thus in my opinion the work is suitable for publication.

We thank the Reviewer no.2 for the comments. We hope to have increased the quality of the paper.

Finally, we had a native English speaking teacher read the revised version of the text, and we made some changes to improve the readability of the manuscript.

Round 2

Reviewer 1 Report

The authors have addressed adequately all of my concerns in the new and revised version of the manuscript.

Line 105 contains a typographical error: "...although KORsuspinal activation..."

Author Response

we thank the Reviewer for pointing out the mistake:

on line 105 of the revised manuscript we amended the mistake putting the word supraspinal, instead of suspinal. sorry for that.